# Reteplase Fc-fusions produced in *N. benthamiana* are able to dissolve blood clots *ex vivo*

**Shiva Izadi[1,2], Mokhtar Jalali Javaran[3]\*, Sajad Rashidi Monfared[3], Alexandra Castilho[1]\***

**1** Department of Applied Genetics and Cell Biology, Natural Resources and Life Sciences, Vienna, Austria, **2** Faculty of Agriculture, Department of Plant Genetics and Breeding, Tarbiat Modares University, Tehran, Iran, **3** Faculty of Agriculture, Department of Agricultural Biotechnology, Tarbiat Modares University, Tehran, Iran

\* m_jalali@modares.ac.ir (MJJ); alexandra.castilho@boku.ac.at (AC)

## Abstract

Thrombolytic and fibrinolytic therapies are effective treatments to dissolve blood clots in stroke therapy. Thrombolytic drugs activate plasminogen to its cleaved form plasmin, a proteolytic enzyme that breaks the crosslinks between fibrin molecules. The FDA-approved human tissue plasminogen activator Reteplase (rPA) is a non-glycosylated protein produced in *E. coli*. rPA is a deletion mutant of the wild-type Alteplase that benefits from an extended plasma half-life, reduced fibrin specificity and the ability to better penetrate into blood clots. Different methods have been proposed to improve the production of rPA. Here we show for the first time the transient expression in *Nicotiana benthamiana* of rPA fused to the immunoglobulin fragment crystallizable (Fc) domain on an IgG1, a strategy commonly used to improve the stability of therapeutic proteins. Despite our success on the expression and purification of dimeric rPA-Fc fusions, protein instability results in high amounts of Fc-derived degradation products. We hypothesize that the "Y"- shape of dimeric Fc fusions cause steric hindrance between protein domains and leads to physical instability. Indeed, mutations of critical residues in the Fc dimerization interface allowed the expression of fully stable rPA monomeric Fc-fusions. The ability of rPA-Fc to convert plasminogen into plasmin was demonstrated by plasminogen zymography and clot lysis assay shows that rPA-Fc is able to dissolve blood clots *ex vivo*. Finally, we addressed concerns with the plant-specific glycosylation by modulating rPA-Fc glycosylation towards serum-like structures including α2,6-sialylated and α1,6-core fucosylated N-glycans completely devoid of plant core fucose and xylose residues.

**Data Availability Statement:** All relevant data are within the paper and its Supporting information files.

## Introduction

Cardiovascular diseases, including acute myocardial infarction (AMI) and ischemic stroke (AIS), are the leading cause of long-term disability [1]. Thrombolytic drugs, particularly tissue plasminogen activators (tPAs), can clear circulatory obstructions due to a fibrin clot or

**Funding:** This work was supported by a grant from the Austrian Science Fund (FWF; https://www.fwf.ac.at/de/): P31131-B32.

**Abbreviations: ADCC**, antibody dependent cellular cytotoxicity; **AF**, apoplastic fluid; **CDC**, complement dependent cytotoxicity; **Fc**, fragment crystallizable domain; **FcRn**, neonatal Fc receptor; **FcγRIIIa**, Fc receptor FcγRIIIa (CD16a); **IgG1**, human immunoglobulin G; **KDa**, kilo Dalton; **LC-ESI-MS**, liquid chromatography-electrospray ionization-tandem mass spectrometry; **mRFP**, monomeric red fluorescent protein; **PA**, plasminogen activator; **rPA**, Reteplase; **tPA**, Alteplase; **TSP**, Total soluble proteins; **ΔXF**, *N. benthamiana* glycosylation mutant plants lacking the plant-specific β1,2-xylose and core α1,3-fucose residues.

thrombus. These fibrinolytic agents are fibrin-specific serine proteases. They work by converting plasminogen to active plasmin, that in turn breaks down the fibrinogen and fibrin in the clot matrix [2] (Fig 1). Different forms of plasminogen activators have been developed to treat thrombotic disease [3,4]. Alteplase (Activase®, Genentech) was the first FDA-approved recombinant tPA identical to native plasminogen activator and is expressed in Chinese hamster ovary (CHO) cells. It is composed of 527 amino acids (68-KDa) and comprises five distinct domains: the finger domain (F) involved in the binding of fibrin and hepatic clearance of tPA [5], the epidermal growth factor-like domain (EGF) which contributes to the hepatic clearance of tPA [6], a kringle 1 (K1) domain which is important in the uptake of tPA by mannose receptors on liver cells [7], a kringle 2 (K2) domain involved in the high-affinity binding to fibrin and activation of plasminogen and a serine protease domain (S) where the catalytic activity of tPA takes place [8] (Fig 1). The zymogen enzyme exists as a single chain that is cleaved (at $Arg^{275}$ in the K2) into a A and a B chain in the presence of plasmin. The A-chain comprises four N-terminal domains and the B-chains include only the serine protease domain where amino acids $His^{322}$, $Asp^{371}$ and $Ser^{478}$ form the catalytic triad that enables the activation of plasminogen to plasmin [9,10].

tPA is a glycosylated protein with four potential glycosites ($Asn^{117}$, $Asn^{184}$, $Asn^{218}$, and $Asn^{448}$). tPA molecules that are glycosylated at position 184 are termed type I while type II molecules are not glycosylated at this position. Glycan analysis of CHO-derived Alteplase revealed that the potential $Asn^{218}$ glycosylation site is not occupied, $Asn^{117}$ carries exclusively high mannose oligosaccharides while $Asn^{184}$ and $Asn^{448}$ carry predominantly di-antennary complex glycans with different amounts of α2,3-terminal sialic acid and core α1,6-fucose, typical of CHO-derived glycoproteins [11,12]. tPA glycosylation was shown to affect its enzymatic activity [13] and its plasma clearance [14]. In particularly, the high-mannose structures at $Asn^{117}$ have been implicated in the rapid clearance from the blood stream through high affinity binding to the mannose receptors [15,16]. The rapid clearance from plasma is also due to the recognition of structural elements on first three N-terminal domains by certain hepatic receptors and has hampered clinical applications of tPA [7,17]. Therefore, through deletions or

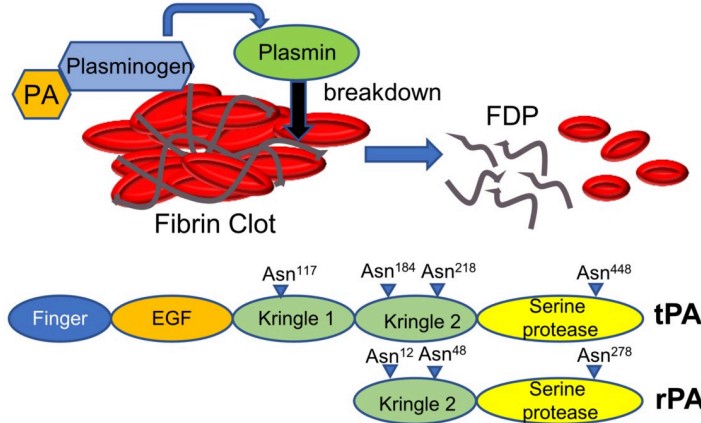

**Fig 1. Schematic representation of tissue plasminogen activators (PA) mode of action and functional domains.** Plasminogen activators cleave the zymogen plasminogen to form the serine protease, plasmin. The fibrinolytic enzyme plasmin breaks the cross-links between fibrin molecules, which are the structural support of the blood clot generating fibrin degradation products (FDP). Alteplase (tPA) has 5 domains: A finger domain, an epidermal growth factor (EGF)-like domain, 2 kringle domains, and a serine protease proteolytic domain. Reteplase (rPA) is a deletion of tPA consisting in only the kringle 2 domain and the serine protease proteolytic domain. Protein glycosylation sites are indicated.

substitutions of the Alteplase sequence, mutants of PAs with diverse pharmacokinetic and pharmacodynamic properties have been developed to treat thrombotic diseases [10,18,19].

Reteplase (rPA) is a 355 amino acid (1–3 and 176–527aa) deletion of tPA that comprises only K2 and S domains of Alteplase with a molecular weight 39-KDa [10,20] (Fig 1). Compared to Alteplase, rPA has a greater thrombolytic potency [21]. Deletions of the F, EGF and K1 domains in rPA result in an extended plasma half-life, reduced fibrin specificity and the ability to better penetrate into blood clots [10,22]. Due to the deletion of the fibronectin finger region, rPA has reduced fibrin binding capabilities. However, increased fibrin affinity has been achieved with a chimeric and mutated rPA [23,24].

rPA has two N-glycosylation sites at $Asn^{12}$ and $Asn^{278}$ (corresponding to $Asn^{184}$ and $Asn^{448}$ in tPA). Glycosylation seems to be involved in regulating rPA protease activity so it is only active in the presence of a fibrin clot [25–27]. It was shown that mutation at $Asn^{12}$ (N12P in K2) results in a delayed clot lysis activity while a faster rate of clot lysis is observed when the $Asn^{278}$ (N278S in S domain) is mutated. In addition, both mutations caused the loss of protease activity. Interestingly, $Asn^{12}$ glycosylation seems to regulate the fibrinolytic activity of rPA since non-glycosylated rPA (double mutants) show the slowest clot lysis activity [28]. FDA-approved Reteplase (Retavase®, Centocor) is a non-glycosylated protein produced in *E. Coli*. Due to the large number of disulfide bonds and lack of glycosylation, rPA produced in *E. coli* cannot be folded properly and is accumulated as inclusion bodies requiring a refolding process [29]. Reteplase has been expressed in other hosts such as *Bacillus* [30]. Yeast [31]., insect [32], mammalian [23,24] and plant cells [33,34]. The expression levels and protein stability vary significantly among all these systems. Since, conventional production methods are inadequate to meet the spiking demands of fibrinolytic proteases, efforts have been undertaken to improve Reteplase production [35]. Recent advances in pharmacological engineering and design revealed the potential of protein fusions for improving the pharmacokinetics of therapeutic proteins. The most widely used approach includes the fusion of the therapeutic protein to the constant fragment (hinge-CH2-CH3, Fc) of a human immunoglobulin G (IgG1) [36]. The Fc domain folds independently and can improve the solubility and stability of the partner molecule [37]. It also provides a number of advantageous biological and pharmacological properties. Indeed, Fc-fusion proteins have a (i) prolonged therapeutic activity due to an increased plasma half-life provided by the neonatal Fc receptor (FcRn) recycling, (ii) slower renal clearance due to higher molecular weight and (iii) reduced immunogenicity [38]. Moreover, from a technological perspective, the Fc region allows for easy cost-effective purification by protein-G/A affinity chromatography [39].

Plants are been used as a platform for the production of pharmaceutical proteins for several years [40]. In particular, the tobacco related plant species, *N. benthamiana*, is well suited for transient expression of recombinant proteins without the need for genetic transformation. Importantly, *N. benthamiana* is highly tolerant to glyco-modifications and recent advances in glyco-engineering facilitate customized *N*-glycosylation of plant-derived glycoproteins [41].

In this investigation we set up to transiently express rPA-Fc fusions in *N. benthamiana* and simultaneously modulate their glycosylation towards human-like structures. We described the production of rPA-Fc fusions decorated with bi-antennary neutral and mammalian-like core fucosylated and α2,6-sialylated glycans. Our results show that rPA-Fc fusions are higly instable and substitution of the hinge region for a flexible stretch of Gly-Ser residues increases the stability of the fusion protein (rPA-$^{L}$Fc). Protein stability can be further improved by fusing rPA to a monomeric Fc (rPA-mFc). Importantly, plant-derived rPA-Fc show *ex vivo* plasma clot lysis activity and *in vitro* proteolytic activity comparable to Alteplase. To our knowledge this is the first report on the production of a tissue plasminogen activator fused to IgG fragments.

This fundamental research provides a baseline for the development of novel thrombolytic drugs with improved pharmacological efficacy.

## Materials and methods

A DNA fragment coding for 355 amino acids (1 to 3 and 176 to 527aa) of rPA (DB00009) lacking the endogenous signal peptide was custom synthesized and codon-optimized for *N. benthamiana*.

The sequence of the heavy-chain of Rituximab IgG1 (HC, Acc. No. AX556949) codon-optimized for *N. benthamiana* was used as template to amplify Fc domains.

### Constructs for rPA gene expression

**Cloning rPA-mRFP.** In order to target rPA to the secretory pathway we substituted the endogenous signal peptide (SP) with the one from barley α-amylase (Acc No CAA33298). To evaluation the subcellular localization, αrPA was fused to the monomeric red fluorescent protein (mRFP). Briefly, the rPA was amplified with primer pair rPA F1/R1 digested with *Xba*I/*Bam*HI and cloned into p31 binary vector (SPα-mRFP, [42] (Fig 2 and S1 Fig).

**Cloning rPA-^HFc.** The primer pair HFc F1/Fc R1 was used to amplify the hinge-CH2-CH3 domains of IgG1 (aa 220–451) flanked by *Eps*3I restriction sites. The fragment was digested with *Eps*3I and cloned into viral-based MagnICON vector pICH26211α [43] digested with *Bsa*I. The resulting cloning vector (pICH26211α: ^HFc) carries the coding sequence for SPα and two internal *Bsa*I sites for further cloning. rPA was amplified with rPA F2/R2 primers, digested with *Bsa*I and cloned into pICH26211α: ^HFc digested the same way. The resulting vector was used to express rPA fused to Fc domain of IgG1 separated by the hinge region (rPA-^HFc, Fig 2 and S1 Fig).

**Cloning rPA-^LFc.** Similarly, the CH2-CH3 domains (aa 239–451) were amplified with primer pair LFc F1/Fc R1 to produce a Fc fragment flanked by *Eps*3I sites and with a (G4S)2 linker and two internal *Bsa*I sites. This fragment was cloned into pICH26211α digested with *Bsa*I resulting in the cloning vector pICH26211α: ^LFc. rPA was amplified using the rPA F2/R3

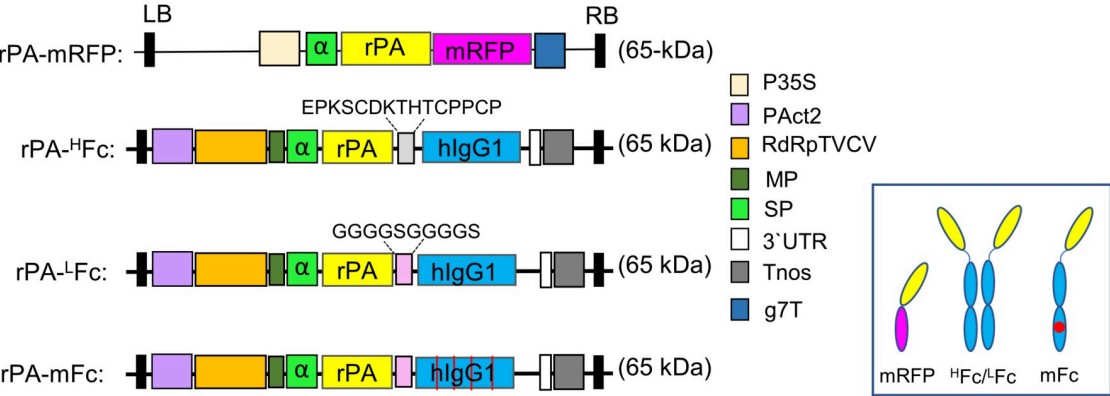

**Fig 2. Schematic representation of the expression vectors used in this investigation.** The binary vector (p31) with a barley α-amylase signal peptide for apoplast targeting was used to clone rPA fused to the monomeric red fluorescent protein (mRFP). The modified TMV-based MagnICON vector (pICHα26211α) was used to clone rPA-Fc-fusion proteins. Common features are: g7T: *Agrobacterium* gene 7 terminator; P35S: Cauliflower mosaic virus (CaMV) 35S promoter; PAct2: Arabidopsis actin 2 promoter; RdRpTVCV: RNA-dependent RNA polymerase from Turnip vein clearing virus; MP: Movement protein from the tobacco mosaic virus; SP: Barley α-amylase signal peptide for apoplast targeting; Tnos: Nopalin synthase gene terminator; 3'UTR: TVCV Pol. 3'-untranslated region; LB: Left border; RB: Right border. The expected protein molecular weight is shown for each fusion (for full peptide sequence see also S1 Fig). Inset show a schematic representation of the oligomerization of the rPA fusions.

primer pair, digested with *Bsa*I and cloned into pICH26211α: $^{LFc}$. The resulting vector was used to express rPA fused to Fc domain of IgG1 separated by a (Gly-Gly-Gly-Gly-Ser)$_2$ flexible linker (rPA-$^L$Fc, Fig 2 and S1 Fig).

**Cloning rPA-mFc.** A monomeric IgG1-$^L$Fc (mFc) was generated by mutating critical residues on the IgG1-Fc dimerization interface located in CH3 domain (T366R, L368H, P395K and F405R) [44,45].

First, a codon optimized DNA fragment coding for a mutated IgG1-CH3 domain (aa 345–451) was custom synthesized. rPA-mFc was generated by overlap extension polymerase chain reaction (OE-PCR). The primer pair rPAF2/mFc R1 was used to amplify the rPA-(G4S)$_2$-CH2 fragment from rPA-$^L$Fc and the mutated CH3 domain was amplified from the synthetic clone with mFc F1/Fc R1 primers. OE-PCR was performed using rPA F2/Fc R1 primers to assemble the rPA-mFc DNA fragment. rPA-mFc was double digested with *Bsa*I/*Eps*3I and cloned into pICH26211α vector digested with *Bsa*I (Fig 2 and S1 Fig).

Primers used for cloning are listed in S1 Table. All recombinant plasmids were transformed into *E. Coli* DH5α. After sequencing, the plasmids were transformed into *Agrobacterium tumefaciens* GV3101.

## Plant material

*N. benthamiana* wild type (WT) and transgenic glycosylation lines ΔXF [46] ΔXF$^{GalT}$ [47] and ΔXF$^{SIA}$ [48] were grown in a growth chamber at 22˚C with a 16 h light/8 h dark photoperiod.

## rPA expression and purification

rPA fusion proteins were transiently expressed in 4- to 5-week-old *N. benthamiana* leaves by agro-infiltration at an optical density at 600 nm (OD600) of 0.1–0.3 as described previously [49]. In some experiments rPA fusions were co-expressed with p19 protein of *Tomato bushy stunt virus* (TBSV) [50].

Total soluble proteins (TSP) and proteins secreted to the apoplastic fluid (AF) were extracted 4–5 days post infiltration with (1:2 w/v) 0.5 M NaCl, 45 mM Tris, 1 mM EDTA, 40 mM ascorbic acid; pH 7 [42].

Quantification of full-length rPA-Fc expression was carried out by immunoblot analysis using anti-IgG antibodies. 10 μL of TSP non-diluted and diluted (1:2; 1:5; 1:20 and 1:100) were loaded on SDS-PAGE and analysed by Western blotting using anti-IgG antibodies. Commercially available IgG1 was used as reference to create a standard curve. Western blot images captured with Fusion Solo S image system (Vilber Lourmat) were analysed using BIO-1D (Vilber Lourmat) software to measure relative band intensities.

rPA-Fc fusions were purified out of TSP by affinity chromatography either with rProteinA or rProteinG (GE Healthcare) according to manufacturer's instructions and protein concentration was determined by NanoDrop$^{TM}$ 2000 spectrophotometer (Thermo Scientific).

## SDS-PAGE and immunoblotting

TSP and purified proteins fractionated by 12% SDS-PAGE under reducing and non-reducing conditions were either stained with Coomassie Brilliant Blue R-250 or transferred to Hybond Enhanced Chemiluminescence nitrocellulose membranes (GE Healthcare) to be analysed by immunoblotting using specific antibodies/lectins.

- anti-hIgG-HRP (1:5000 in 5% Milk/PBS-Tween, Promega)

- mouse polyclonal anti-human tPA antibody (produced at Tarbiat Modares University, Iran) detected with HRP-conjugated anti-mouse-IgG.

- anti-Lewis-A antibodies (1:400 JIM84 in 3% Milk/PBS-Tween, provided by Richard Strasser, BOKU, Vienna), detected using HRP-conjugated anti-rat IgM (Jackson, 1:10000 in PBS-Tween).

- Biotinylated *Ricinus Communis Aggutinnin* I lectin (RCA, Vector, 5–10 μg/mL in 3% BSA-biotin free/PBS-Tween), detected using HRP-conjugated Streptavidin (1mg/mL, Vector, 1:8000 in PBS-Tween).

- Biotinylated *Sambucus Nigra* lectin (SNA, Vector, 5μg/mL in 10mM Hepes; 0.15M NaCl; 0.1M CaCl2; 0.04%NaN3, pH7.5), detected using HRP-conjugated Streptavidin.

Clarity™ Western enhanced chemiluminescence reagents (Bio-Rad, Life Science) were used as substrates. Western blot images captured with Fusion Solo S image system (Vilber Lourmat). Finally, membranes blots were stained with Ponceau S (Sigma Aldrich) to visualize transferred proteins. Activase® (Alteplase or tPA,s Genentech) as used as a control.

## Subcellular localization

rPA-mRFP was transiently co-expressed with a plasma membrane marker EGFP-LTI6b (PM-GFP, [42]) and subcellular localization was determined by live-cell confocal laser scanning microscopy (CLSM). High-resolution images were acquired 2 days post infiltration (dpi) on an upright Leica SP5 II confocal microscope using the Leica LAS AF software system [51]. Dual-color imaging of cells expressing both GFP and mRFP was performed simultaneously using a 488-nm argon laser line and the 543-nm helium/neon laser line. GFP emission was recorded at 500-535nm, whereas mRFP fluorescence was detected at 600-630nm. Post-acquisition image processing was performed in Affinity photo.

## *In vitro* degradation assays

Plant derived rPA-$^L$Fc was incubated with apoplastic fluid, concentrated (10x) by ultrafiltration using Amicon YM30 centrifugal filter units (Merck Millipore, Burlington, MA, USA). After 1-16h incubation, samples were analysed by immunoblotting with anti-IgG antibodies.

## Analysis of glycosylation status

Purified rPA-Fc was deglycosylated with Endoglycosidase H (Endo H) and with N-glycosidase F (PGNaseF) (New England Biolabs) for 2 h at 37˚C following the manufacturer's directions. Treatments in absence of the glycosidase were used as controls. Samples were analysed by Western blotting with anti-hIgG-HRP antibodies.

## Site-specific *N*-glycoprofiling and total glycans analysis

Site-specific *N*-glycoprofiling of Alteplase (tPA) and rPA-Fc proteins was determined by mass spectrometry using liquid chromatography-electrospray ionization-tandem (LC-ESI-MS) analysis, as described previously [52]. Briefly, protein bands were excised from SDS-PAGE gels, S-alkylated with iodoacetamide and proteolytically digested with trypsin to generate glycopeptides. The peptides were separated using a nanoEase C18 column (nanoEase M/Z HSS T3 Column, 100Å, 1.8 μm, 300 μm X 150 mm, Waters) using 0.1% formic acid as the aqueous solvent. A gradient from 3.5% B (B: 80% ACN, 20% A) to 40% B in 40 min was applied, followed by a 5 min gradient from 40% B to 95% B that facilitates elution of large peptides, at a flow rate of 6 μL/min. Detection was performed with an QTOF MS (maXis 4G, Bruker) equipped with the standard ESI source in positive ion, DDA mode.

The possible glycopeptides were identified as sets of peaks consisting of the peptide moiety and the attached N-glycan varying in the number of HexNAc units, hexose, deoxyhexose and pentose residues. The theoretical masses of these glycopeptides were determined with a spread sheet using the monoisotopic masses for amino acids and monosaccharides.

For total glycan analysis, the glycopeptides were digested with PGNaseF and the released N-glycan were characterized by MALDI-MS and porous graphitic carbon (PGC)-LC-E-SI-MS) [52].

### Plasma clot assay

The proteolytic activity of plant-derived rPA-Fc was evaluated by fibrin clot assay. The effect of the plant-derived rPA-Fc and tPA on plasma fibrinolysis was assessed by *ex vivo* plasma clot-based assay. Briefly, 10 mL of blood plasma (Iranian Blood Transfusion Organization) clotted with 25 mM calcium chloride were poured on Petri dishes and purified rPA-Fc was spotted on the fibrin plate at different concentration and incubated at 37˚C overnight. Activase® (Alteplase or tPA,s Genentech) as used as a control.

### Plasminogen zymography

Zymography was performed to evaluate the activity of the plant-derived rPA-Fc. Plasminogen zymography is based on the principal that when plasminogen and gelatin are incorporated into the polyacrylamide matrix, tPA and rPA will activate plasminogen to solubilize gelatin [53]. Purified plant-derived rPA-Fc fusions were fractionated in 8% SDS-PAGE resolving gels supplemented with 0.1% (w/v) gelatin (Sigma, USA) and plasminogen (10µg/mL CoaChrom Diagnostica GmbH). Electrophoresis was run at a continual current of 10 mA at 4˚C. SDS was remove from the gel by washing sequentially twice in 2.5% Triton X-100, followed by rinsing in 0.1 M Glycine/NaOH pH 9.3 for 6h at 37˚C. After staining with Coomassie Brilliant Blue R-250 and distaining, areas of enzyme activity appear as clear bands against a blue background. Activase® (Alteplase or tPA, Genentech) and a human IgG1 monoclonal antibody were used as positive and negative controls, respectively.

### Ethics statement

Experiments with blood plasma were conducted according the ethical code standards as defined by the ethical Committee in Research at the Tarbit Modares University from whom we obtained informed written consent.

## Results

### Subcellular localization of rPA-mRFP

The signal peptide of the barley alpha amylase (SPα) was used in this investigation to substitute the rPA endogenous SP and the target recombinant protein to the secretory pathway. To confirm the secretion of αrPA, the protein was C-terminally tagged with the monomeric red fluorescent protein (rPA-mRFP, Fig 2 and S1 Fig) and transiently expressed it in *N. benthamiana* leaves by agroinfiltration. The subcellular localization of rPA-mRFP was analyzed by live-cell confocal microscopy at 2dpi. Co-expression of rPA-mRFP with a plasma membrane-localized fusion protein (GFP–LTI6b) shows significant co-localization of the fluorescent protein fusions (Fig 3), typical of secreted proteins.

**Fig 3. Subcellular localization of rPA-mRFP.** Subcellular localization of rPA-mRFP was determined by live-cell imaging of infiltrated *N. benthamiana* leaf epidermal cells. rPA-mRFP was transiently co-expressed in *N. benthamiana* with EGFP-LTI6b (PM-GFP), a plasma membrane marker, and analyzed at 2dpi. The fluorescence (magenta) signal is typical of secreted proteins. Superposed images (merged) show significant co-localization of rPA-mRFP and PM-GFP. Scale bar is indicated.

## Expression of dimeric rPA-Fc fusions

The magnICON® minimum-virus system has been widely used for efficient large-scale production of several recombinant proteins. The system is extremely effective for the rapid and high-yield production of several pharmaceutical proteins. Here we used the virus-vector based on turnip vein clearing Tobamovirus (TVCV, pICH26211α [43]) to express rPA fused to the hinge-CH2-CH3 region of an IgG1 (rPA-HFc, Fig 2 and S1 Fig).

Total soluble proteins (TSP) were extracted from infiltrated leaves 4–5 days post infiltration (dpi) and analysed by Western blotting with anti-hIgG antibodies. The full-length fusion protein is express as a ~72-kDa protein. The higher molecular weight of the fusion protein when compared to the calculated 65-kDa is most probably due to protein glycosylation (see below). In addition to full-length rPA-HFc two smaller molecular weight bands are detected at 40- and 34-kDa (Fig 4A), indicating instability of the fusion protein. Co-expression of the RNA silencing suppressor p19, known to increase levels of transient expression, significantly boosted the expression of full-length rPA-HFc (Fig 4A). Nevertheless, the rPA-HFc fusion is highly unstable and protein purification out of TSP mainly yields 34-kDa free Fc (S2 Fig). Similar results have been reported by us and others indicating that the hinge region of monoclonal antibodies is prone to proteolytic degradation [54]. Other than linking the two Fc domains, the hinge region is not important for Fc function but introduces potential instability due to disulfide reduction, and also via enzymatic degradation at several protease cleavage sites. Alternative flexible peptide linkers are used to improve protein folding and have shown increasing importance in the construction of stable fusion proteins [55]. Here, we designed a construct to express rPA and Fc domains separated by a flexible linker consisting of a stretch of Glycine (Gly) and Serine (Ser) residues ("GS" linker). The (Gly-Gly-Gly-Gly-Ser)$_2$ sequence was inserted between the rPA and CH2-CH3 domains to provide flexibility and allow for mobility of the connecting domains (rPA-LFc, Fig 2 and S1 Fig). TSP and apoplastic fluids (AF, representing the plant secretome) collected from infiltrated leaves were analysed by immunoblotting with anti-IgG antibodies (Fig 4B). Contrasting with non-fused Fc (LFc, S3 Fig), the rPA-LFc full-length fusion (~72-kDa) does not accumulate in the AF. These results indicate that either the fusion protein does not cross the cell wall or when it does it is degraded by proteases in the apoplastic fluid. Acidic pH mimics the conditions of AF *in planta* [56] where several proteases are active [57]. To investigate if rPA-LFc can be degraded by AF proteases *in vitro*, we incubated the purified protein with acidic AF (pH 5.0) isolated from wild-type *N. benthamiana* plants. Samples were analysed at several time points with IgG antibodies. The results show that the full-length rPA-LFc (72-kDa) is still detected after 3h incubation but is completely degraded after 8h. No

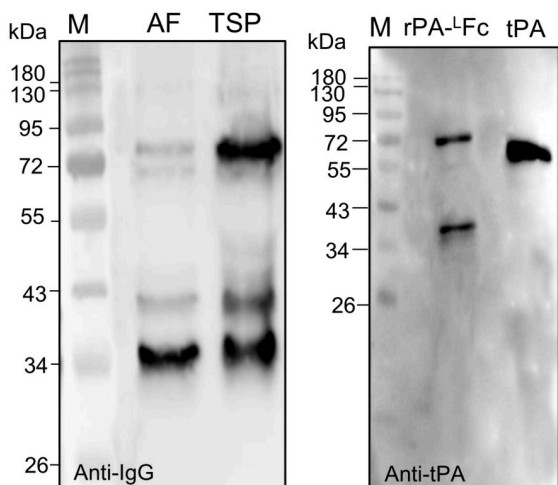

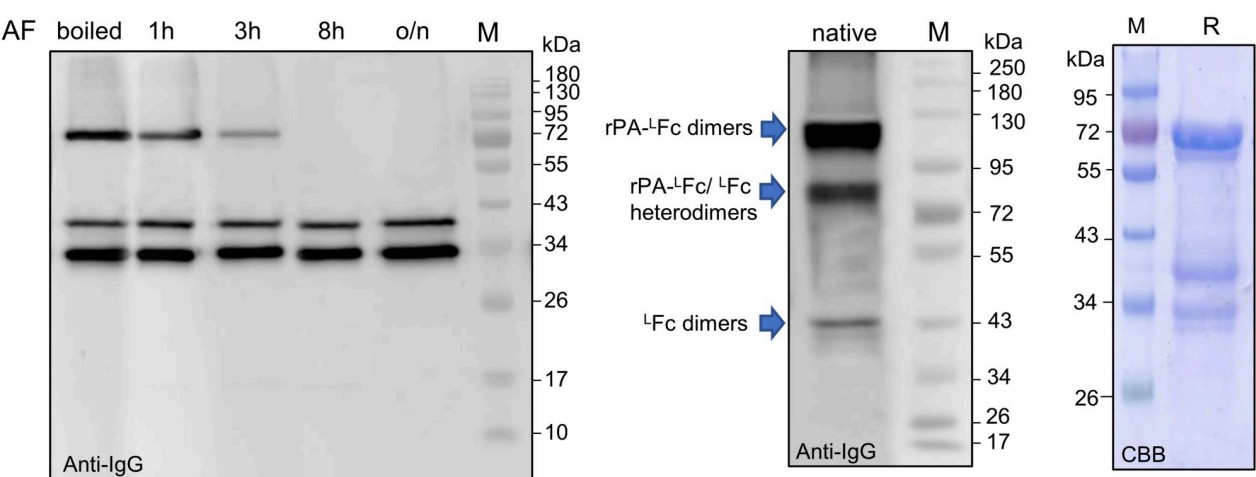

**Fig 4. Expression of dimeric rPA-Fc fusions in *N. benthamiana*. (A)** Total soluble proteins extracts (TSP) from *N. benthamiana* ΔXF leaves expressing rPA-HFc co-expressed with (+) or without (-) p19 were analysed in 12% SDS-PAGE under reducing conditions with anti-hIgG antibodies. (**B**) TSP and apoplastic fluids (AF) from *N. benthamiana* ΔXF leaves expressing rPA-LFc were analysed in 12% SDS-PAGE under reducing conditions with anti-hIgG antibodies (left panel). rPA-LFc and tPA (Alteplase) were also analysed with anti-tPA antibodies (right panel). (**C**) Protein A purified rPA-LFc was incubated with acidic AF isolated from *N. benthamiana* wild type plants. Samples were analysed at different time points with anti-hIgG antibodies. Incubation with AF previously boiled to inactivate plant proteases served as a negative control. (**D**) TSP from *N. benthamiana* ΔXF leaves expressing rPA-LFc were analysed in 8% native PAGE with anti-hIgG antibodies (left panel) and protein A purified rPA-LFc was stained with Commassie brilliant blue under reducing conditions (CBB) (right panel). Protein size markers are shown in kilo Dalton (kDa).

degradation was visible when rPA-LFc was incubated with AF previously heated to 95˚C to inactivate proteases (Fig 4C).

Next, we expressed rPA-LFc with or without p19 and collected TSP at 5, 7 and 9 dpi. Contrasting to rPA-HFc, no significant impact of p19 on the expression rPA-LFc which show similar levels from 5–9 dpi (S4A Fig). The expression level of full-length rPA-LFc on TSP was

estimated at 0.2 mg/g of fresh leaf material (S4B Fig). Although the expression of full-length rPA-$^L$Fc is significantly higher than rPA-$^H$Fc, the protein is still unstable and immunoblotting with anti-IgG antibodies detects three protein bands. These bands are also visible after protein A purification in commassie gels under reducing conditions and represent the full-size (~72kDa), the N-terminus truncated fusion (40-kDa) and the free Fc (~34-kDa) (Fig 4D). The 72- and 40-kDa bands are also detected with anti-tPA antibodies but the 34-kDa band reacts only to anti-IgG antibodies (Fig 4B). Protein cleavage was also observed for tPA [33,58] and rPA expressed in *Nicotiana* species [34]. In general, tPA and rPA expressed in eukaryote cells can be hydrolysed (R-I bond in K2, S1 Fig) into A and B-chains by plasmin, trypsin and other serine proteases [33,58]. The 40-kDa band that reacts to both anti-tPA and anti-IgG antibodies should consist of free Fc (34-KDa) with ~6-kDa from the C-terminus of rPA. This is not in accordance with hydrolysis to generate A and B chains. The expression of rPA-$^L$Fc analysed in native conditions shows three bands representing rPA-$^L$Fc/rPA-$^L$Fc homodimers (~120-kDa), Fc/Fc homodimers (~43-kDa), and rPA-$^L$Fc/Fc heterodimers(~80-kDa). The discrepancy of these molecular weights to the calculated mass for dimers is probably due to protein folding (Fig 4D). Indeed, analysis of the expression of non-fused $^L$Fc shows the presence of 34-KDa in reducing samples and of 43-kDa in native conditions (S3 Fig). Accordingly, dimers of truncated rPA-$^L$Fc (~40-kDa) should have a molecular weight slightly higher than Fc dimers and therefore we speculate that the 80-kDa band represents rPA-$^L$Fc/Fc heterodimers and not truncated rPA-$^L$Fc dimers. One possible explanation is that cleavage of rPA-$^L$Fc to produce a ~40-kDa truncated protein arises from sample preparation in reducing conditions (reducing disulphide bridges and protein denaturation). Similar results were observed when strep-tagged rPA was expressed in *N. benthamiana* and degradation of the 45-kDa protein to 31- and 14-kDa products was only visible when the sample was analysed under reducing conditions [34].

## Expression of monomeric rPA-Fc fusions

Production of Fc fusions in plants often results in significant amounts of degradation products [54,59,60]. These molecules spontaneously fold and dimerize to mimic the "Y"-shape of a full IgG without interactions with a light chain [61]. To further investigate the importance of these protein-protein interactions, we expressed the heavy chain of the anti-ebola 3F6 monoclonal antibody [62] without the corresponding light chain. After protein A purification the samples were analysed by Western blotting using anti-heavy chain antibodies. Full-length heavy chain was detected as expected at 55-kDa but a ~34-kDa fragment reacting with anti-heavy chain antibodies was also detected. Degradation was not detected when the heavy chain of 13F6 was co-expressed with the light chain (S5 Fig).

To address the instability of the rPA-$^H$Fc and rPA-$^L$Fc fusions, we hypothesize that the dimeric "Y"-shape configuration creates steric hindrance between the rPA domains leading to physical instability. Therefore, we aimed for the expression of rPA fused to an immunoglobulin domain that is not able to form dimers.

Fc dimerization of hingeless Fc fusions (e.g. rPA-$^L$Fc) is mediated mainly by a large hydrophobic interface in its CH3 domain, involving at least 16 residues in each polypeptide chain that make intermolecular interactions [63]. Disruption of this large interaction by extensive mutations in the CH3 domain allowed the identification Fc proteins that are >99% monomeric (mFc) and carry only 4 mutations (S351L, T366R, L368H, P395K) [44,45,64].

Here we set up to express a mFc by introducing four mutations (T366R, L368H, P395K and F405R) in the CH3 domain of $^L$Fc. We have excluded mutation at S351L and introduce a F405R mutation based on follow-up studies that showed S351L mutation was involved in the

binding of mFc to unrelated proteins [45] and that F405 mutated to R, Q or E favored Fc monomer formation [65].

Transient expression of the mFc in *N. benthamiana* was analysed under reducing and non-reducing conditions. Contrasting with dimeric [L]Fc (S3 Fig), immunoblotting with anti-IgG antibodies reveal the presence of a ~34-kDa band in reducing and at ~26-kDa in native PAGE (Fig 5A), indicating that the protein is no longer able to dimerize.

According to this result we generated a fusion of rPA to mFc (rPA-mFc, Fig 2 and S1 Fig). rPA-mFc was transiently expressed in the presence or absence of p19 and TSP were analysed with anti-IgG antibodies at 4-, 5- and 7-dpi. As observed for rPA-[H]Fc, p19 drastically increases the expression of rPA-mFc but no significant differences are observed for the expression of rPA-mFc between 4–7 dpi (Fig 5B). rPA-mFc is mainly detected as a double band (~72-kDa) under reducing conditions (Fig 5B) with an expression level estimated at 0.18 mg/g of fresh leaf material (S6 Fig). To investigate if the double band observed for full-length rPA-mFc is due to protein underglycosylation we co-express rPA-mFc with LmSTT3D, a single-subunit oligosaccharyltransferase from the protozoan *Leishmania major* [66]. Analysis of TSP with anti-IgG antibodies clearly shows a single band upon co-expression of OST (Fig 5C).

rPA-mFc was purified out of TSP with protein G since it was shown that mFc has an increased binding affinity to protein G compared to protein A [64]. In contrast to dimeric rPA-[L]Fc, purified rPA-mFc shows a single protein band at the expected size for the full-length monomeric protein (~72-kDa) (Fig 5D). Analysis of purified rPA-mFc in reducing and non-reducing conditions shows no dimerization of rPA-mFc. The non-reduced sample has a slightly lower molecular size (~60-Ka) due to protein folding. Also, a faint band at ~45-kDa might represent a different protein conformation (Fig 5D).

## Glyco-engineering of plant-derived rPA

Reteplase is a glycoprotein with three potential N-glycosylation sites at $Asn^{12}$, $Asn^{48}$, and $Asn^{278}$ (corresponding to $Asn^{184}$, $Asn^{218}$ and $Asn^{448}$ in tPA). In both tPA and rPA, glycosite $Asn^{218}$ ($Asn^{48}$ in rPA) is not glycosylated and $Asn^{184}$ ($Asn^{12}$ in rPA) is only 20–25% occupied [11,12].

To assess the glycosylation status of plant-derived rPA-[L]Fc, we first treated the purified protein with Endo H and PNGaseF. A clear shift in mobility was detected after PNGase F digestion while no shift was visible upon Endo H digestion, suggesting that rPA-[L]Fc is mainly decorated with complex-type glycans (Fig 6A).

In order to modulate the glycosylation profile, we transiently expressed rPA-[L]Fc in *N. benthamiana* wild type (WT) and in glycosylation transgenic lines lacking the plant typical xylose and fucose core epitopes (ΔXF, [46]); ΔXF overexpressing the human β1,4-galactosyl-transferase (ΔXF[GalT], [47]); and in ΔXF overexpressing the entire protein sialylation pathway (ΔXF[SIA], [48]). The presence of particular glycoforms was first evaluated by immunoblotting with JIM84 (for Lewis A epitopes), RCA (for terminal β1,4-galactose) and SNA (for terminal α2,6-sialic acid). Fig 6B shows that Lewis-A epitopes are only detected in WT-rPA-[L]Fc and to a lesser extent in ΔXF-rPA-[L]Fc. Lewis-A epitopes are drastically reduced or abolished in rPA-[L]Fc expressed in ΔXF[GalT] and in ΔXF[SIA] most probably due to competition of the β1,3- and β1,4-galactosyltransferases for the same substrate. Terminal human-like β1,4-galactosyla-tion is detected in ΔXF[GalT]-rPA-[L]Fc and in ΔXF[SIA]-rPA-[L]Fc while protein sialylation is detected only in ΔXF[SIA]-rPA-[L]Fc. SNA lectins preferentially bind to terminal sialic acid in a α2,6-linkage and to a lesser degree in a α2,3-linkage therefore the sialylation signal of CHO-derived tPA is weaker. SNA lectin blots also detected sialylation in rPA-mFc expressed in ΔXF[SIA] plants.

## A. Expression of mFc

## B. rPA-mFc time-course expression

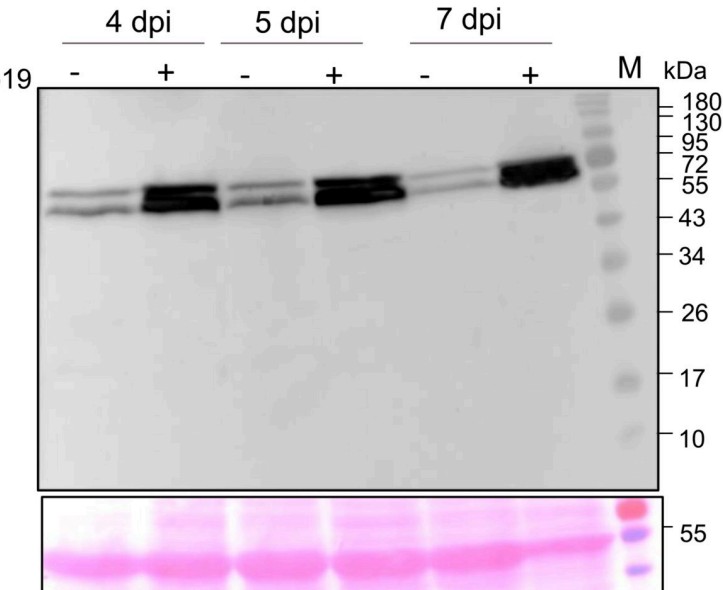

## C. rPA-mFc glycosylation status

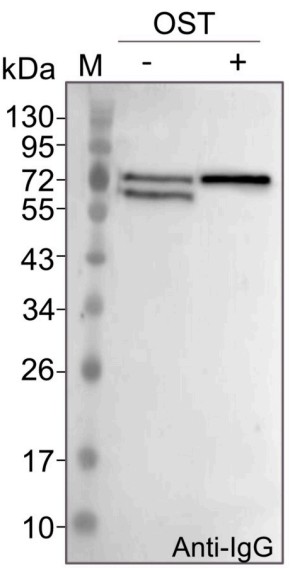

## D. rPA-mFc purification and dimerization

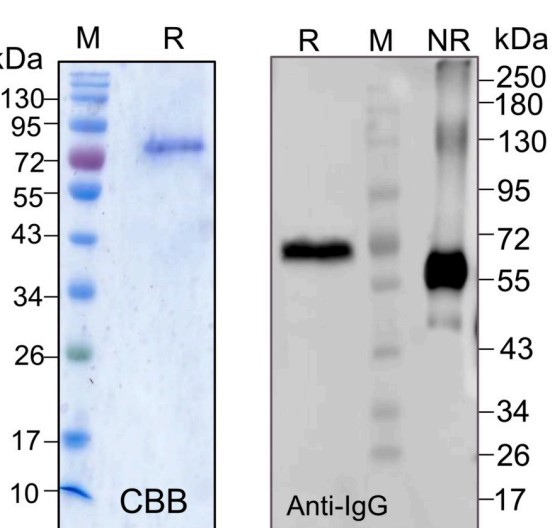

**Fig 5. Expression of monomeric rPA-Fc fusions in *N. benthamiana*.** (A) Total soluble proteins extracts (TSP) from *N. benthamiana* ΔXF leaves expressing mFc (non-fused) were analysed in 12% SDS-PAGE under reducing (R) and non-reducing (NR) conditions and in native PAGE with anti-hIgG antibodies. (B) TSP from *N. benthamiana* ΔXF leaves expressing rPA-mFc co-expressed with (+) or without (-) p19 were analysed in 12% SDS-PAGE under reducing conditions with anti-hIgG antibodies at different days post infiltration (dpi). Ponceau staining shows similar amounts of protein loaded. (C) rPA-mFc was expressed in *N. benthamiana* ΔXF with (+) or without (-) co-expression of an oligosaccharyltransferase from *Leishmania major* (OST). TSP were analysed in 12% SDS-PAGE under reducing conditions with anti-hIgG antibodies. (D) rPA-mFc purified out of TSP with Protein G was stained with Commassie brilliant blue under reducing conditions (CBB) (left panel) and analysed in 8% SDS-PAGE under reducing (R) and non-reducing (NR) conditions with anti-hIgG antibodies (right panel). Protein size markers are shown in kilo Dalton (kDa).

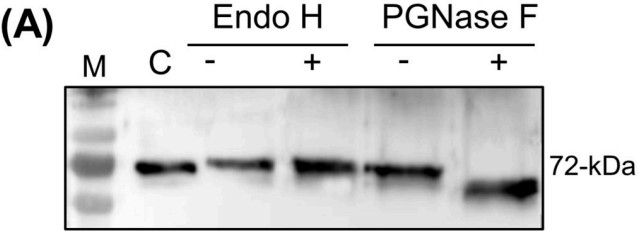

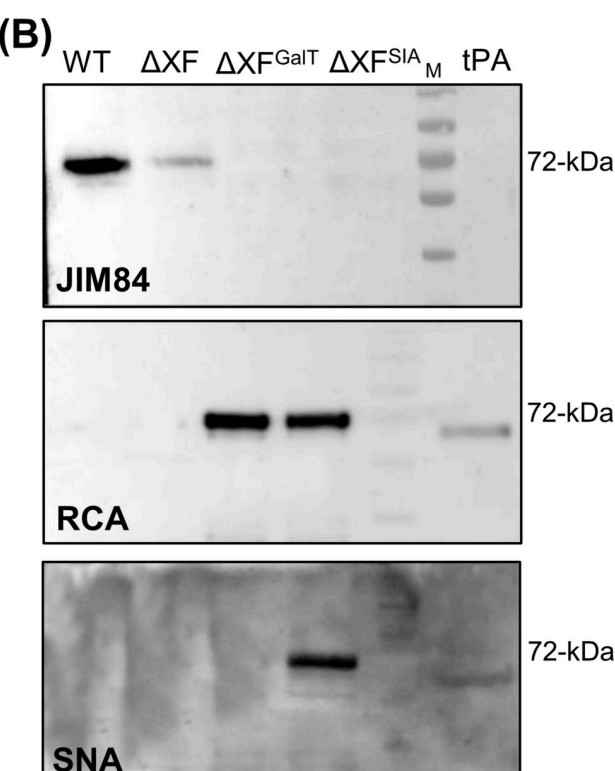

**Fig 6. Glycosylation status of plant-derived rPA-$^L$Fc. (A)** Purified rPA-$^L$Fc expressed in *N. benthamiana* ΔXF plants was digested with Endo H and PNGase F and analysed with anti-hIgG antibodies. Undigested rPA-$^L$Fc served as control (C). **(B)** Purified rPA-$^L$Fc expressed in *N. benthamiana* wild type (WT), in ΔXF; in ΔXF overexpressing the human β1,4-galactosyltransferase (ΔXF$^{GalT}$) and in ΔXF overexpressing the entire protein sialylation pathway (ΔXF$^{SIA}$) were analysed for the presence of Lewis-A epitopes (JIM84), terminal β1,4-galactose (RCA) and terminal α2,6-sialic acid (SNA). tPA (Alteplase) was used as a control.

Finally, we characterize the N-glycosylation pattern in a site-specific manner. Alteplase and rPA-Fc expressed in different glycosylation plant hosts were digested with trypsin and the glycopeptides were subjected to LC-ESI-MS analysis. N-glycans could be detected on 3 sites of Alteplase (tPA, Asn$^{117}$, Asn$^{184}$ and Asn$^{448}$). As expected, glycosylation at Asn$^{117}$ consists of high mannose glycans with Man5 being the main glycoform (S7A Fig) while Asn$^{184}$ and Asn$^{448}$ carry core fucosylated bi- tri- and tetra-sialylated glycans (Fig 7A). Asn$^{448}$ is fully glycosylated (non-glycosylated peptide was not detected) while Asn$^{184}$ is only 26% glycosylated. Glycosylation profile of plant-derived rPA-$^L$Fc is similar on both glycosites (Asn$^{12}$ and Asn$^{278}$). The glycosylation profile of ΔXF-rPA-$^L$Fc is highly homogenous with almost a single glycoform depleted of plant specific core xylose and fucose residues (84–89% GnGn, Fig 7B).

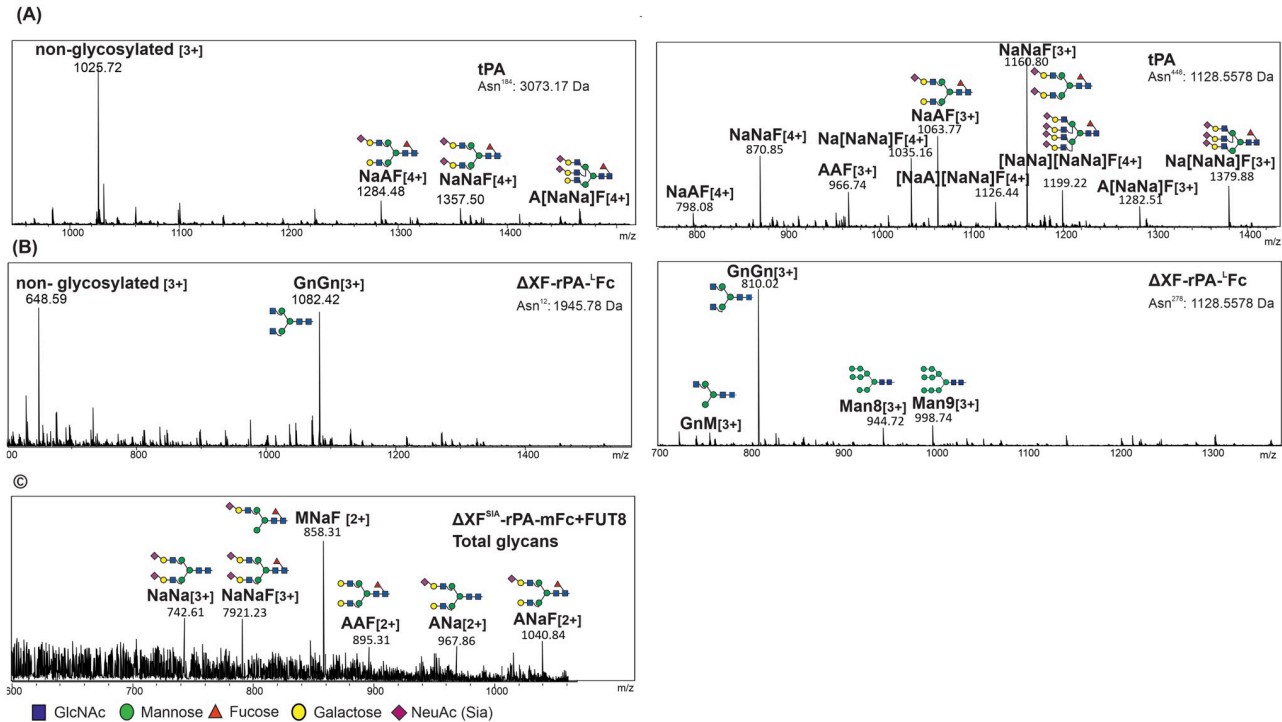

**Fig 7. Glycan analysis of CHO-derived tPA and plant-derived rPA-Fc fusions.** Tryptic glycopeptides of tPA (Alteplase) and ΔXF-rPA-$^L$Fc were analysed by LC-ESI-MS. N-glycosylation profiles ([M+3H]3+ and [M+4H]4+) are shown for tPA glycosites Asn[184]: YSSEFCSTPACSEG**N**SDCYFG NGSAYR and Asn[448]: CTSQHLL**N**R and for ΔXF-rPA-$^L$Fc glycosites Asn[12]: SYQG**N**SDCYFGNGSAYR and Asn[278]: CTSQHLL**N**R. Glycosylation of tPA at Asn[117] glycosite is shown in S6B Fig. Total glycans released from ΔXF$^{SIA}$-rPA-mFc co-expressed with mammalian α1,6-fucosyltransferase (FUT8) were analysed by MALDI-MS. The major glycosylated peaks are depicted. The assigned N-glycan structures were labelled according to the ProGlycAn nomenclature (http://www.proglycan.com/). A cartoon illustration highlights the main glycan structure detected for each peptide (http://www.functionalglycomics.org/).

Lewis A structures detected with JIM84 immunoblotting were not detected by mass spectrometry. Asn[12] (Asn[184] in tPA) is 56% glycosylated and Asn[278] (Asn[448] in tPA) is fully glycosylated. Glycosylation was also analysed for rPA-$^L$Fc and rPA-mFc co-expressed in ΔXF$^{SIA}$ plants with mammalian core-fucosylation (FUT8, α1,6-fucosyltransferase). Total glycans released with PNGaseF were analysed by MALDI-MS. MS results show similar profiles for rPA-LFc and rpA-mFc with mainly mono-sialylated and core fucosylated glycans (MNaF) with bi-sialylated glycans detected to a lesser extend (NaNa and NaNaF) (Fig 7C). It should be noticed that total glycan analysis also includes glycans released from the glycosite located in the Fc domain (Asn[297]) which are known to be poorly sialylated [67]. Incomplete sialylation of plant-derived proteins (MNaF) have been previously reported and seem to be due to partially inactivation of the Golgi β-1,2-N-acetylglucosaminyltransferase II (GnTII). We anticipate that over-expression of GnTII should increase sialylation of rPA-Fc [43].

## Biological activity of plant-derived rPA-Fc fusions

The clot lysis assay was used to determine the proteolytic activity of plant-derived rPA-$^L$Fc. Unlike the fibrin plate assay, this assay uses macroscopic blood clots thereby better mimicking *in vivo* biology. Grey spots or solubilized plasma, representing the digestion of fibrin polymers, were detected in the areas spotted by rPA-$^L$Fc expressed in ΔXF plants and commercial tPA

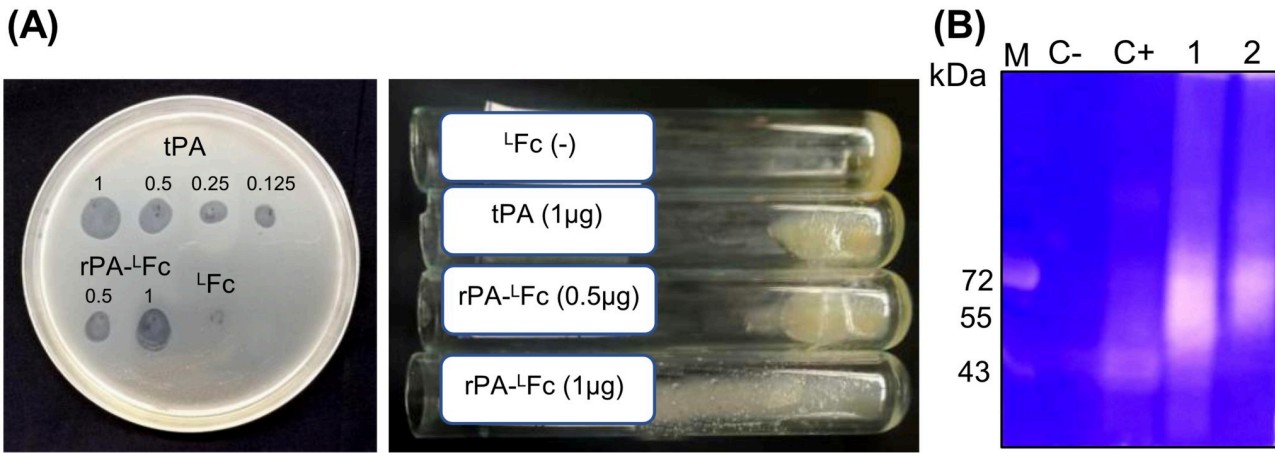

**Fig 8. Functional assays of rPA-Fc fusions expressed in ΔXF plants. (A)** The proteolytic activity of rPA-<sup>L</sup>Fc was assessed by clot lysis assay. Clotted blood plasma was poured on Petri dishes (left) or in glass tubes (right) and purified rPA-<sup>L</sup>Fc was spotted at different concentrations (μg). tPA (Alteplase) was used as a positive control and Fc as a negative control. Fibrin digestion can be visualized by grey spots (left panel) or solubilized plasma (right panel). **(B)** Plasminogen zymography of purified plant-derived rPA-Fc. Staining of 8% SDS-PAGE with Coomassie brilliant blue shows the proteolytically cleaved sites as white clear smears on a dark blue background. 1. rPA-mFc and 2. rPA-<sup>L</sup>Fc. Proteolytic activity is also observed on the positive control, (C+, Alteplase) but not on the negative control (C-, IgG1). Protein size markers are shown in kilo Dalton (kDa).

but not on Fc spots (negative control) (Fig 8A). The results show that rPA-<sup>L</sup>Fc is active e*x-vivo* in a dose dependent manner comparable to Alteplase.

The plasminogenolytic activity of plant-derived rpA-Fc (rPA-<sup>L</sup>Fc and rPA-mFc) was also evaluated on the basis of a zymography. rPA-Fc activity was analysed in SDS-PAGE containing copolymerized plasminogen and gelatin. The idea is that plasminogen and gelatin, incorporated into the polyacrylamide matrix at the time of casting, serve as *in situ* substrates for the localization of plasminogen activator bands by negative staining. 3μg of purified non-reducing samples of rPA-Fc (rPA-<sup>L</sup>Fc and rPA-mFc) were loaded on 8% SDS-PAGE gel. Alteplase and IgG1 monoclanal antibody (4 μg) were used as positive and negative controls, respectively.

After staining with Coomassie brilliant blue we observe the presence of a white smear for rPA-<sup>L</sup>Fc and rPA-mFc showing their proteolytically activity. For alteplase, a more discrete band around 40-kDa is observed and most probably indicate the activity of two chain's tPA (chain A: 31-kDa and chain B: 28-kDa). The discrepancy in molecular size is due to glycosylation. Importantly, there is no evidence of proteolytic activity in the negative control, IgG1 (Fig 8B). Compared to alteplase, the activity of plant-derived rPA-Fc is detected as broadened bands (smears) and therefore it is hard to distinguish between one or two chain's proteins. Expanded reactive bands of proteolytic enzymes were observed previously [68,69].

## Discussion

Reteplase (rPA) is a deletion variant of Alteplase (tPA) retaining two functional regions of the protease (thrombolytic Kringle2 and protease domain) as well as the N-terminal 1–3 amino acid residues. Compared to native tPA, rPA has longer plasma half-life (14–18 vs. 3–4 min), better diffusion in the clot and higher fibrinolytic activity [10,22].

Recombinant expression of tPA and rPA in plants have used different approaches including stable expression in *Laminaria japonica* [70], in Cucumis species [71,72] or in *N. tabacum* [58,73–75] and transient expression in *N. benthamiana* [34]. In these studies, plant-derived rPA (or tPA) tagged with strepII or His tags was targeted to the chloroplast, ER, cytoplasm and apoplast.

*N. benthamiana* is most suited for the rapid large-scale synthesis of recombinant proteins and recent advances in plant glycoengineering have allowed the production of plant-derived protein with tailored human-like glycosylation [41]. Here we described for the first time the transient expression of rPA fused to the Fc region of an IgG1 monoclonal antibody (rPA-Fc). rPA and Fc domains were separated either by the hinge region of the IgG1 (rPA-$^H$Fc) or by a flexible (Gly4Ser)$_2$ linker (rPA-$^L$Fc) to ensure correct protein folding and function. Transient expression of rPA-$^H$Fc and rPA-$^L$Fc in *N. benthamiana* showed that the fusion proteins are highly instable and a large fraction of the recombinant protein consists of Fc lacking the rPA fragment (free Fc). Many other therapeutically important proteins have been expressed in different plant species fused to a Fc domain and to our knowledge the majority of these studies have reported on the presence of degradation products derived from the Fc partner expressed [76–83].

Fc-fusions inherit some antibody-like properties such as bivalency. Indeed rPA-Fc are able to form dimers that mimic the "Y-shape" of an antibody but lack the interactions with the light chain. Expression of the heavy chain of an IgG1 without co-expression of the correspondent light chain showed similar degradation. We hypothesize that in this dimeric configuration, rPA domains are adjacent to each other and steric hindrance between protein domains might lead to physical instability. Besides the disulfide bridges between hinge regions (not present in rPA-$^L$Fc), Fc dimerization is also mediated by a large hydrophobic interface in its CH3 domain, which involves at least 16 residues in each polypeptide chain [45]. Here we have introduced four mutations on the CH3 domain of rPA-$^L$Fc to generate a monomeric fusion (rPA-mFc). Transient expression of rPA-mFc show that the protein is not able to dimerize. Importantly, rPA-mFc is stable and no significant protein degradation was observed. These results seem to indicate that avoiding dimerization of the fusion Fc partner can be one strategy to produce stable fusions proteins.

In addition, mFc fusions can be exploited to greatly expand the traditional Fc therapeutic applications, since the mFc (i) has a smaller size (ii) binds to FcRn and exhibits a similar *in vivo* half-life to that of dimeric Fc; (iii) lacks binding to FcγRIIIa resulting in the absence of Fc-mediated cytotoxicity including ADCC and CDC, and therefore can serve as a safe, non-toxic platform for the delivery of prolonged half-life to therapeutic proteins [64].

Expression level of full-length rPA-$^L$Fc and rPA-mFc fusions in TSP are approximately 0.2 mg/g and the fusion proteins can be efficiently purified by immunoaffinity with yields up to 70μg/g of fresh leaf. These expression levels are significantly higher than those reported earlier using the same expression system [34] and similar to those of rPA expressed in tobacco chloroplast [73]. Chloroplast transformation is an attractive strategy to achieve high levels of protein expression but lacks the machinery for glycosylation. Most studies on plant-derived rPA did not consider the glycosylation status of the recombinant protein. rPA carries two out of the three glycosylation sites of tPA (Asn$^{12}$ and Asn$^{278}$). It was initial described that non-glycosylated tPA variants have higher fibrinolytic versus fibrinogenolytic ratio compared to fully glycosylated counterparts [25]. However, later it was shown that both residues are critical to the *in vitro* biological activity of reteplase [28]. The host cell line used to produce a glycoprotein has a strong influence on its glycosylation since different host systems have varying repertoire of glycosylation enzymes that contributes to heterogeneity in glycosylation profiles. Despite their wide use in the production of human recombinant proteins, non-human mammalian cells lack or have differing machineries for human-type glycosylation. For example, CHO cells do not express α2,6-sialyltransferases and therefore sialic acids found in CHO-derived glycoproteins are exclusively α2,3-linked in contrast to human proteins which contain mainly α2,6-linked sialic acids [84]. Also, proteins expressed in CHO and in human cell lines (e.g.

HEK) show significant differences in glycostructures, including a consistently increased number of sialic acids probably due to branching of glycans [85].

So far, plant produced rPA is either non-glycosylated (chloroplasts), decorated with high mannosidic (ER retention) or complex glycans with plant specific epitopes (core β1,2-xylose and α1,3-fucose). These can lead to aggregation/miss folding of non-glycosylated rPA, shorter half-life due to binding to mannose receptors and mannose binding proteins or immunogenic reactions owing to plant typical glycosylation.

Here we successfully modulated the glycosylation of plant-derived rPA-Fc fusions towards serum-like glyco-structures. Glycosylation of rPA-Fc expressed in ΔXF plants is extremely homogenous and devoid of plant specific epitopes. rPA is up to 56% glycosylated at $Asn^{12}$ which is significantly higher than tPA (26% at $Asn^{278}$) even without co-expression of OST. Importantly, we show that rPA-Fc glycosylation can be modulated to carry α2,6-sialylated glycans with human-like core α1,6-fucose, thus mimicking the profile of the majority of serum glycoproteins. These results and our previous achievements in plant glycoengineering [41] encourage future studies on the impact of specific glycoforms on the therapeutic efficacy of rPA.

Finally, the biological activity of plant-derived rPA-Fc was demonstrated by plasminogen zymography and clot lysis assay, methods commonly designed to assess the *in vitro* and *ex vivo* activity of PAs in converting plasminogen to plasmin.

Altogether, our results on the successful stabilization, glycoengineering and biological function of plant-derived rPA-Fc fusions is encouraging but additional efforts should be followed to optimize the expression levels.

## Supporting information

**S1 Fig. Protein sequence of Alteplase (tPA), reteplase (rPA) and rPA fusions.** Green: SP; Magenta: mRFP; Blue: CH2-CH3 domain of Fc; Grey: Linker. Potential N-glycosylation sites are in blue, mutations on CH3 domain are in red. Hydrolysis of the PA peptide bond between Arg-Ile (RI, in green) in the kringle-2 domain generates two polypeptide chains.
(PDF)

**S2 Fig. Protein A purified rPA-**$^{H}$**Fc stained with Commassie brilliant blue (CBB).** Protein size markers are shown in kilo Dalton (kDa).
(TIFF)

**S3 Fig. Expression of $^{L}$Fc.** Total soluble proteins extracts (TSP) and apoplastic fluid (AF) from *N. benthamiana* leaves expressing $^{L}$Fc (fusion partner) were analysed by immunoblotting with anti-IgG antibodies (left panel). TSP were also analysed under reducing and non-reducing conditions with anti-IgG antibodies (right panel). Protein size markers are shown in kilo Dalton (kDa).
(TIFF)

**S4 Fig. Time course and quantification of rPA-$^{L}$Fc expression. a** Total soluble proteins extracts (TSP) from *N. benthamiana* leaves expressing rPA-$^{L}$Fc with (+) or without (-) co-expression of p19 were analysed by immunoblotting with anti-IgG antibodies at 5-, 7- and 9-days post infiltration (dpi). **b** Expression of full-length rPA-$^{L}$Fc (72-kDa) in TSP was estimated by a semi-quantitative western blot analysis with anti-IgG antibodies using IgG1 as standard. Protein size markers are shown in kilo Dalton (kDa).
(TIFF)

**S5 Fig. Expression of 13F6 monoclonal antibody.** Total soluble proteins extracts (TSP) from *N. benthamiana* leaves expressing the heavy chain (HC) of the 13F6 monoclonal antibody

were analysed by immunoblotting with anti-γ chain antibodies (HC of IgG1). Full 13F6 antibodies (HC and LC) purified out of TSP with protein A were either stained with Commassie brilliant blue (CBB) or subjected to immunoblotting with anti-γ chain antibodies. Protein size markers are shown in kilo Dalton (kDa).
(TIFF)

**S6 Fig. Quantification of rPA-mFc expression.** Expression of rPA-mFc (72-kDa) in TSP was estimated by a semi-quantitative western blot analysis with anti-IgG antibodies using IgG1 as standard. Protein size markers are shown in kilo Dalton (kDa).
(TIFF)

**S7 Fig. Glyco-profilling of Alteplase Asn[117].** N-glycosylation ([M+3H]3+ and [M+4H]4+) of tPA glycosite Asn[117] (GTWSTAESGAECTNW**N**SSALAQKPYSGR) was analysed by LC-E-SI-MS. The major glycosylated peaks are depicted. The assigned N-glycan structures were labelled according to the ProGlycAn nomenclature (http://www.proglycan.com/). A cartoon illustration highlights the main glycan structure detected for each peptide (http://www.functionalglycomics.org/).
(TIFF)

**S1 Table. Primers used in PCR for gene cloning and construction of rPA fusion variants.** *Xba*I (TCTAGA) *Bam*HI (GGATCC) *Bsa*I (GGTCTC) *Eps*3I (CGTCTC) restriction sites are in bold. Single-strand DNA *overhangs* are underlined.
(PDF)

**S1 Raw images.**
(PDF)

## Acknowledgments

We thank Clemens Gruber from the BOKU Core Facility Mass Spectrometry (CFMS) for the glycan analysis. We also thank colleagues from Icon Genetics GmbH, Germany for providing access to their magnICON® expression system and Dr. Manouchehr Mirshahi, Iran for help with the Plasma clot lysis experiments. We would like to thank the Ministry of Sciences, Research and Technology, Biotechnology Development Council in Iran and the Austria Agency for Education and Internationalisation (Ernst Mack Grant worldwide) for supporting Mss Izadi.

## Author Contributions

**Conceptualization:** Alexandra Castilho.

**Funding acquisition:** Alexandra Castilho.

**Investigation:** Shiva Izadi, Alexandra Castilho.

**Methodology:** Shiva Izadi.

**Project administration:** Alexandra Castilho.

**Supervision:** Mokhtar Jalali Javaran, Sajad Rashidi Monfared, Alexandra Castilho.

**Writing – original draft:** Shiva Izadi, Alexandra Castilho.

**Writing – review & editing:** Alexandra Castilho.

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
