## [Decision Letter · Decision Letter 0]

10 Nov 2021

PONE-D-21-29875Reteplase Fc-fusions produced in N. benthamiana are able to dissolve blood clots ex vivoPLOS ONE

Dear Dr. Castilho,

Thank you for submitting your manuscript to PLOS ONE. After careful consideration, we feel that it has merit but does not fully meet PLOS ONE’s publication criteria as it currently stands. Therefore, we invite you to submit a revised version of the manuscript that addresses the points raised during the review process.

We look forward to receiving your revised manuscript.

Kind regards,

Arijit Biswas

Academic Editor

PLOS ONE

Journal Requirements:

2. Please update your submission to use the PLOS LaTeX template. The template and more information on our requirements for LaTeX submissions can be found at http://journals.plos.org/plosone/s/latex

In your cover letter, please note whether your blot/gel image data are in Supporting Information or posted at a public data repository, provide the repository URL if relevant, and provide specific details as to which raw blot/gel images, if any, are not available. Email us at plosone@plos.org if you have any questions

4. We note that you have included the phrase “data not shown” in your manuscript. Unfortunately, this does not meet our data sharing requirements. PLOS does not permit references to inaccessible data. We require that authors provide all relevant data within the paper, Supporting Information files, or in an acceptable, public repository. Please add a citation to support this phrase or upload the data that corresponds with these findings to a stable repository (such as Figshare or Dryad) and provide and URLs, DOIs, or accession numbers that may be used to access these data. Or, if the data are not a core part of the research being presented in your study, we ask that you remove the phrase that refers to these data

Reviewers' comments:

Reviewer's Responses to Questions

**Comments to the Author**

1. Is the manuscript technically sound, and do the data support the conclusions?

Reviewer #1: Yes

Reviewer #2: Yes

2. Has the statistical analysis been performed appropriately and rigorously? 

Reviewer #1: Yes

Reviewer #2: N/A

3. Have the authors made all data underlying the findings in their manuscript fully available?

Reviewer #1: Yes

Reviewer #2: Yes

4. Is the manuscript presented in an intelligible fashion and written in standard English?

Reviewer #1: Yes

Reviewer #2: Yes

5. Review Comments to the Author

Reviewer #1: This work described transient expression of rPA fused to the Fc region of an IgG1 monoclonal antibody (rPA-Fc) either by the hinge 464 region of the IgG1 (rPA-HFc) or by a flexible (Gly4Ser) linker (rPA-LFc) to ensure correct protein folding and function. The authors showed that point mutations in the CH3 domain of rPA-LFc reduced degradation through inhibition of protein dimerization. The rPA-FC produced recombinantly showed accepptable purity and high activity regarding the clot lysis assay.

The paper is written in a logical order and is understandable. The images are well designed with acceptable quality and resolution, and the legends of figures are enough informative.

There are some major and minor points which can be considered to improve the data evaluations:

Major concerns:

In general, some of the assays used (SDS-PAGE and immunoblotting, in vitro degradation assay, and Plasma clot assay) are all semi-quantitative although there are various quantitative assays in this regards. If the authors have access to the systems7assay kits, it is highly recommended that some of the assays suggested below will be used as complementary assays to confirm the activity of rPA.

1. Plate-based activity assay using PA-specific chromogenic/fluorogenic substrates might be used to evaluate the activity of recombinantly produced rPA. This assay can be used not only to evaluate the biological activity of rPA, but also to monitor the degradation of rPA.

2. The rotational thromboelastometry (ROTEM) can be used to evaluate clot lysis assay by spiking citrate pool plasma/whole blood with the rPA and the clot formation and lysis can be monitored. In this way the lysis parameters such as lysis onset time (LOT) and maximum lysis (ML) will be helpful to quantify clot lysis.

Minor concern:

1. In the introduction section, Nicotiana benthamiana was repeatedly mentioned as Nicotiana benthamiana or N. benthamiana. After the first introduction it might be mentioned constantly as N. benthamian.

Reviewer #2: This is a well written comprehensive report on the biochemical aspects in production of rPA-FC fusions constructs. The presented results are of general interest beyond rPA-Fc-specific aspects and may therefore be published in PLOS ONE. However, I have some comments as stated below.

Major:

The construct finally used for the clot lysis assay (rPA-LFC) is only specified in Fig. 8 but not within the corresponding results section. Why wasn't rPA-mFC also tested using this assay? Which glycosylation variant was used for assessing the biological activities?

The paper would benefit from more physiological experiments, e.g. whole blood ROTEM / TEG analysis.

It appears that the discussion does not match the level of the described results (the first paragraph reads like an introduction and some parts of rest like a summary). During revision, it may reasonable to transfer some aspects already adressed within the results section.

Minor:

There are some typos within the manuscript, please review carefully. For example: Lines: 283/284, 291/292, 350/351, 361, 431, 470

Fig. 2. It would be helpful to also include schematic images on the different structures of the constructs (dimeric, monomeric).

The resolutions of Fig. 7 and the lower panel of Fig. 5B need to be increased.

6. PLOS authors have the option to publish the peer review history of their article (what does this mean?). If published, this will include your full peer review and any attached files.

Reviewer #1: **Yes: **Nasim Shahidi Hamedani

Reviewer #2: No

---

## [Author Response · Author response to Decision Letter 0]

15 Nov 2021

Reviewer #1: 

This work described transient expression of rPA fused to the Fc region of an IgG1 monoclonal antibody (rPA-Fc) either by the hinge 464 region of the IgG1 (rPA-HFc) or by a flexible (Gly4Ser) linker (rPA-LFc) to ensure correct protein folding and function. The authors showed that point mutations in the CH3 domain of rPA-LFc reduced degradation through inhibition of protein dimerization. The rPA-FC produced recombinantly showed accepptable purity and high activity regarding the clot lysis assay.

The paper is written in a logical order and is understandable. The images are well designed with acceptable quality and resolution, and the legends of figures are enough informative.

There are some major and minor points which can be considered to improve the data evaluations:

Major concerns:

In general, some of the assays used (SDS-PAGE and immunoblotting, in vitro degradation assay, and Plasma clot assay) are all semi-quantitative although there are various quantitative assays in this regards. If the authors have access to the systems7assay kits, it is highly recommended that some of the assays suggested below will be used as complementary assays to confirm the activity of rPA.

1. Plate-based activity assay using PA-specific chromogenic/fluorogenic substrates might be used to evaluate the activity of recombinantly produced rPA. This assay can be used not only to evaluate the biological activity of rPA, but also to monitor the degradation of rPA.

2. The rotational thromboelastometry (ROTEM) can be used to evaluate clot lysis assay by spiking citrate pool plasma/whole blood with the rPA and the clot formation and lysis can be monitored. In this way the lysis parameters such as lysis onset time (LOT) and maximum lysis (ML) will be helpful to quantify clot lysis.

Reply: We agree with the reviewer that additional biological assays would improve the manuscript. The aim of our investigation was to set up the basic studies and lay the groundwork for the production of rPA-Fc proteins with added value. We took advantage of the short stay of the PhD student (Ms Izadi) in Iran to carry out simple experiments showing that the plant-derived rPA-Fc is active in vitro and ex vivo. She did indeed try to perform a rotational thromboelastometry (ROTEM) assay but encounter technical problems (even the positive control did not work) and run out of time to optimized the method. At BOKU we are a plant-oriented lab and do not have the facilities or permission to perform experiences involving blood/plasma. We however envision the possibility of making available our proteins to other researchers for further studies in future collaborations. 

Minor concern:

1. In the introduction section, Nicotiana benthamiana was repeatedly mentioned as Nicotiana benthamiana or N. benthamiana. After the first introduction it might be mentioned constantly as N. benthamiana.

Reply: We have modified the manuscript accordingly

Reviewer #2: 

This is a well written comprehensive report on the biochemical aspects in production of rPA-FC fusions constructs. The presented results are of general interest beyond rPA-Fc-specific aspects and may therefore be published in PLOS ONE. However, I have some comments as stated below.

Major:

The construct finally used for the clot lysis assay (rPA-LFC) is only specified in Fig. 8 but not within the corresponding results section. Why wasn't rPA-mFC also tested using this assay? Which glycosylation variant was used for assessing the biological activities?

The paper would benefit from more physiological experiments, e.g. whole blood ROTEM / TEG analysis.

Reply: The clot lysis assays were performed at Tarbiat Modares University during a short period visit from Mss Izadi (PhD student). At the moment we were still trying to overcome the issue of instability and the rPA-mFc construct was not yet available for testing. The rPA-mFc construct was used in the zymogram and performed similarly to rPA-LFc. 

The biological activity of rPA-LFc and rPA-mFc was assessed with variants produced in ∆XF plants. This information is now included in the manuscript.

We agree with the reviewer that additional biological assays would improve the manuscript. However, the main aim of our investigation was to set up the basic studies and lay the groundwork for the production of rPA-Fc proteins with added value. We have performed basic experiments to show that our protein is active in vitro and ex vivo. During a short time visit to Tarbiat Modares University, Ms Izadi also tried to perform a rotational thromboelastometry (ROTEM) assay but encounter technical problems (even the positive control did not work) and run out of time to optimized the method. At BOKU, we are a plant-oriented lab and do not have the facilities or permission to perform experiences involving blood/plasma. We however envision the possibility of making available our proteins to other researchers for further studies in future collaborations. 

It appears that the discussion does not match the level of the described results (the first paragraph reads like an introduction and some parts of rest like a summary). During revision, it may reasonable to transfer some aspects already addressed within the results section.

Reply: We agree with the reviewer and modified the discussion section accordingly

Minor:

There are some typos within the manuscript, please review carefully. For example: Lines: 283/284, 291/292, 350/351, 361, 431, 470

Reply: We have modified the manuscript to correct typos

Fig. 2. It would be helpful to also include schematic images on the different structures of the constructs (dimeric, monomeric).

Reply: We have included schematic representation of the oligomerization of the constructs in Figure 2.

The resolutions of Fig. 7 and the lower panel of Fig. 5B need to be increased.

Reply: We have improved resolution of figures

---

## [Editor Report · Decision Letter 1]

17 Nov 2021

Reteplase Fc-fusions produced in N. benthamiana are able to dissolve blood clots ex vivo

PONE-D-21-29875R1

Dear Dr. Castilho,

We’re pleased to inform you that your manuscript has been judged scientifically suitable for publication and will be formally accepted for publication once it meets all outstanding technical requirements.

Kind regards,

Arijit Biswas

Academic Editor

PLOS ONE

---

## [Editor Report · Acceptance letter]

18 Nov 2021

PONE-D-21-29875R1 

Reteplase Fc-fusions produced in N. *benthamiana * are able to dissolve blood clots *ex vivo *

Dear Dr. Castilho:

I'm pleased to inform you that your manuscript has been deemed suitable for publication in PLOS ONE. Congratulations! Your manuscript is now with our production department. 

Kind regards, 

on behalf of

Dr. Arijit Biswas 

Academic Editor

PLOS ONE